# A Review of the Major Prosthetic Factors Influencing the Prognosis of Implant Prosthodontics

**DOI:** 10.3390/jcm10040816

**Published:** 2021-02-17

**Authors:** Javier Montero

**Affiliations:** Department of Surgery, University of Salamanca, 37007 Salamanca, Spain; javimont@usal.es; Tel.: +34-923-294-500 (ext. 1996)

**Keywords:** prognosis, prosthodontics, dental implant, clinical performance, risk factor

## Abstract

Background: The treatment plan of prosthetic restorations supported by dental implants requires comprehensive scientific knowledge to deliver prostheses with good prognosis, even before the implant insertion. This review aims to analyze the main prosthetic determinants of the prognosis of implant-supported prostheses. Methods: A comprehensive review of the literature was conducted with a PICO (Patient Intervention Comparison Outcomes) question: “For partially or complete edentulous subjects treated with implant-supported prostheses, which prosthetic factors could affect clinical outcomes?”. A literature search was performed electronically in PubMed (MEDLINE), Scopus and Cochrane Library with the following equation [PROGNOS * OR RISK] FACTOR IMPLANT DENTAL, and by hand search in relevant journals and throughout the selected papers. Results: This revision was carried out based on 50 papers focused on several prosthodontics-related risk factors that were grouped as follows: implant-connection, loading protocol, transmucosal abutments, prosthetic fit, provisionalization, type of retention, impression technique, fabrication technique, and occlusion. More than a half of the studies were systematic reviews (30%), meta-analysis (16%), or prospective evaluations of prosthesis with various kinds of events (18%). However, narrative reviews of literature (14%) and in vitro/animal studies (16%) were also found. Conclusions: The current literature provides insufficient evidence for most of the investigated topics. However, based on the accumulated data, it seems reasonable to defend that the best treatment approach is the use of morse taper implants with transmucosal abutments, recorded by means of rigidly splinted copings through the pick-up technique, and screwed by milled prosthesis occlusally adjusted to minimize functional overloading.

## 1. Introduction

Edentulism continues to be an oral health challenge with a growing interest, especially due to the increase in life expectancy in industrialized countries; therefore, in the last decades, implant prosthetics have become exponentially popular in both adult and elderly populations.

This growth in demand has forced prosthodontics, as a dental specialty, to be systematically challenged by the incessant changes in the conception and therapeutic approach of dental implants, as well as by the abundance of materials and manufacturing methods available today for the replacement of lost tissues in the stomatological area. An example of this is the thoroughness with which the first implants came onto the market (after 10 years of monitoring of clinical trials and background publications) and the speed with which new materials, new prosthetic additments, and new manufacturing methods emerge on the market nowadays (sometimes without having gone through the minimum scientific forum of clinical effectiveness testing).

Therefore, many of the clinical protocols that we apply today are still based on expert opinion and theories of biological plausibility, rather than on a scientific basis supported by the accumulation of experimental evidence. If we intend to apply evidence-based dentistry, we must place at the service of the patient those clinical procedures in which we are well trained, which have been endorsed by the best scientific evidence and which also satisfy the patient’s preferences or perceptions.

On top of that, there are sometimes results in the literature which, although statistically significant, might be clinically irrelevant or even counterproductive if the patient’s perception of the effects of the various interventions was systematically recorded. For example, although there is a statistically significant difference in the average periodontal insertion gain of 0.5 mm of a regenerative periodontal surgery treatment (test) vs. a conventional ultrasonic prophylaxis (control), surely none of the operators would undergo such an intervention to obtain such a meagre result. In many cases, we should record and learn from qualitative results (which dimensions of well-being worsened or improved after an intervention) rather than quantitative results (ISQ (Implant Stability Quotient) values, mm, µm, BIC (Bone Implant Contact)).

Since the success criteria of the restorative clinician (anatomical and technical complexity of the case, achieved control of the oral disease, precision of the prosthetic and occlusal adjustment, esthetics of the rehabilitation, biomechanics of the rehabilitation, etc.) are far from the success criteria of the patient (number of treatment appointments and touch-ups, waiting time, duration of the therapeutic phase, money invested, hygienic requirements and perception of the result with respect to esthetical-functional expectations) it would be advisable to record this complementary information for an overall view of the therapeutic effect.

In any case, clinicians and researchers need to monitor the influence of prognostic factors on the clinical or subjective outcome of treatments. Given the technical and logistical complexity of prognostic factor studies, there are 6 times more papers in the scientific literature that focus on evaluating clinical outcomes of dental implants (using the keywords Clinical Outcomes Dental Implant) rather than prognostic factors (using [Prognos * OR Risk] Factor Dental Implant) as searched in March 2020. The Glossary of Prosthodontics Terms [1] defines prognosis as “a forecast as to the probable result of a disease or a course of therapy”.

A risk factor is a variable associated or correlated with a negative event (disease or infection), but not necessarily causal. To establish causality, the principle of temporal precedence of the cause is required, which only occurs in longitudinal studies. Ideally, risk factors are assessed by comparing the occurrence of an event in exposed vs. unexposed groups to that potentially effect-modulating variable. Since implant loss is a rare event in the short to medium term, it may be of interest to the clinicians to focus on the occurrence of some type of biological or mechanical complication of the treatment.

According to the Glossary of Oral and Maxillofacial Implants [2], a complication is an “unexpected deviation from the normal treatment outcome”. Usually, data are provided on the complication rate per 100 years (event rate per 100 years), which counts the observation time and the number of implants inserted or prosthesis evaluated. For example, 100 prostheses observed for 1 year each, with only one complication, would have an event rate of 1 per 100 years.

There are two types of complications that must be distinguished: biological and prosthetic. Biological complications are those negative events that affect the tissues surrounding the implant substructure (presence of pain, morbidity, infection, suppuration, mobility, bone resorption, dysesthesia, etc.). Prosthetic complications are negative events that affect the exoprosthesis (meso or superstructure), either to the prefabricated components (mechanical complications) or to the covering material made by the technician (technical complications) [3]. There is positive feedback between biological and prosthetic complications which is bidirectional, that is to say, prosthetic complications can lead to biological complications and vice versa. Among the biological complications, the most feared by clinicians is peri-implantitis, due to its difficult management. Thanks to the American Academy of Periodontology, we now know that the etiopathogenetic path of peri-implantitis has shown to be triggered by bacterial infection that activates a cytokines cascade leading to inflammatory bone loss [4]. This is the reason for considering the stability of the marginal bone tissue around the fixtures as one of the most important clinical parameters in determining the success of a fixed prosthetic rehabilitation.

However, technical complications are considerably more frequent than biological events, as reported in the current literature [5,6,7,8]. In particular, the fracture of acrylic resin teeth [9] or veneering material [5] is quite common. In a recently published paper retrospectively analyzing the clinical performance of 225 rough-surface (plasma-sprayed) titanium implants with at least 25 years of function, a cumulative time-dependent gradient of complications was observed. Specifically, the cumulative incidences of peri-implantitis at 10, 15, and 25 years were 15.3%, 21.0%, and 27.9%, respectively. The cumulative survival rates of mechanical complication-free prostheses at 10, 15, and 25 years were 74.9%, 68.8%, and 56.4%, respectively [10]. Most likely, some of these complications could be explained by prosthodontic, biomechanical, and occlusal factors rather than by factors inherent to the patients themselves. In this sense, Canullo et al. have recently pointed out that the odds ratio of peri-implantitis in healthy implants increases by 48.2 due to implant malposition, 18.7 due to occlusal overload and 3.7 due to the occurrence of some technical complication with the prosthesis [11].

However, to the extent of our knowledge, there are few studies that address the impact of prosthodontic, biomechanical, and occlusal factors on the prognosis of implant prosthetics [12,13]. The objective of this study was to analyze, through a systematic review, the most important prosthetic determinants that could influence the prognosis of implant prosthodontics.

## 2. Materials and Methods

A comprehensive review, according to PRISMA guidelines, was performed in Pubmed, Scopus and Cochrane Library by using the following equation [PROGNOS * OR RISK] FACTOR IMPLANT DENTAL to answer the following PICO (Patient Intervention Comparison Outcomes) question: “For partially or complete edentulous subjects treated with implant-supported prostheses, which prosthetic factors could affect clinical outcomes?” There was no time limitation. The last search was made on 2 April 2020. After reviewing the abstracts of those papers potentially related to the focus of the study (*n* = 360), the full-text articles were then reviewed (*n* = 82). Based on these readings, other relevant papers (*n* = 32) were collected by manual searching within their bibliography to complement the primary core of articles (*n* = 18). The flow chart of the searching steps is shown in Figure 1. After reviewing the final core of full-text papers (*n* = 50), all the prognosis factors were identified and grouped accordingly in order to provide a simplified but panoramic explanation and discussion of the main determinants of the clinical performance of implant prostheses. The search strategy used in this study has given preference to recent systematic reviews, meta-analysis, and randomized clinical trials, therefore the number of papers supporting each factor was not exhaustively included, having discarded some relevant studies whose research focus has been addressed by a more updated or better designed study.

The level of the evidence of each article was established by the study design as good (meta-analysis, systematic review), medium (clinical trial, cohort studies, retrospective studies), and low (narrative review, animal studies and in vitro studies). An effort has been made to compute the quality within each type of study. For Systematic Reviews the quality was computed if the literature search was performed on at least two databases, the scientific quality of the included studies was properly assessed and documented, and the methods used to combine or summarize the findings of studies were appropriate. Similarly, for Clinical Trials the quality was computed if the intervention was randomized and concealed, and the sample size greater was than 30 participants followed for more than one year. For narrative reviews, the quality was assessed depending on the description of the PICO question; the hierarchy of evidence used, and the clinical relevance of the findings.

No statistical calculations were made to summarize or aggregate the study results because of the lack of homogeneity of the study designs. By contrast, this comprehensive review will assess the scientific basis of the available evidence supporting each prosthetic issue and qualitatively summarize the key factors that would potentially affect the prognosis of rehabilitation. A summary index of the level of evidence supporting each topic (Summary Scientific Score = SSS) was calculated as a summation of the number of papers multiplied by the level of evidence (where good was arbitrarily coded as 10; medium, as 1; and low, as 0.1). Then, if a topic was supported by only 1 meta-analysis, it would reach the same score as if it was supported by 10 medium-level studies or 100 low-level studies.

## 3. Results

Prosthodontic treatment on dental implants has continuously evolved over the last decades and is currently a routine option for clinicians and patients. In the field literature, most of the reported prognostic factors of dental implants are focused on evaluating those variables that impinge on the primary osseointegration [14,15,16]. However, the success of the implants depends on the stability of the marginal bone when the prosthesis is connected to them and in function [17]; thus, the potential weight of the prosthetic factors on the prognosis is evident. In fact, current conception of implant success also includes prosthetic parameters and even patient-focused variables such as satisfaction [18], in addition to the traditional implant and peri-implant level criteria.

Therefore, this revision was carried out based on 50 papers (Figure 1) focused on several prosthodontics-related risk factors that were grouped into the following topics: implant-connection, loading protocol, interface integration, provisionalization, type of retention, impression technique, fabrication technique, biomechanics, and occlusion. The distribution of the study design of such papers is shown in Figure 2. More than a half of the studies were systematic reviews (30%), meta-analysis (16%), or prospective evaluations of prosthesis with various kinds of events (18%). Furthermore, narrative reviews of the literature (14%), and in vitro/animal studies (16%) were selected.

Despite the weaknesses in the level of evidence for most prosthodontics topics, it was decided to approach such factors with awareness of limitations of the current knowledge or statements in order to share useful insights with clinicians and researchers. In an attempt to quantify the scientific basis of each topic, the summary scientific score (SSS) endorsing each topic will be commented in the appropriate section (Figure 3).

### 3.1. Prosthetic Complications Rates

Despite the high survival rate of implant-supported prostheses and substantial improvements in implant dentistry over time, esthetic, biologic, and technical complications are still frequent and even not totally avoidable [8]. According to the meta-analysis of Pjetursson, conventional tooth-supported prostheses had a significantly higher 5-year success rate (84.3%) than implant-supported FDPs (Fixed Dental Prostheses) (61.3%), whereas the cantilever tooth-supported FDPs remained in an intermediate position (79.4%) [7]. Furthermore, a systematic review found that the fracture of both the prosthetic screw and the veneering material are common prosthetic complications with fixed implant-supported complete-arch each 5 years of function (10.4% and 33.3%, respectively) [19].

As for single crowns, another meta-analysis by Pjetursson has recently reported that the cumulative 5-year technical complication rates ranged between 8.3–17.5%, whereas the biological complications were less common and ranged between 2.8–8.1% during this period [20].

### 3.2. Effect of the Implant-Connection

According to recent literature [21], the external hexagon does not prevent the formation of a micro gap between the implant and the abutment, which would cause technical and biological complications, mainly when this connection is subjected to high occlusal loads. By contrast, internal connection implants have a greater contact area between the implant and the abutment, allowing better load dissipation along the axis and providing greater joint stability. However, only the morse taper connection provides close contact between implant and abutment, thus achieving the best antibacterial seal and subsequently best marginal bone stability [21,22].

However, even though so far there are no randomized clinical trials addressing the clinical performance of the internal connection vs. the external connection [23], the summary scientific score was still high (SSS = 30), since there were 3 well-performed systematic reviews supporting the abovementioned statements.

### 3.3. Effect of Loading Time

This is probably one of the simplest factors for randomized testing in a dental clinical trial. Thus, there is strong evidence to meta-analyze data [24,25,26,27] and support the following statements: Immediate loading could be performed in implants if the operator is properly trained on implant-prosthodontics protocols and if implants are preferably self-tapping, microroughed, properly sized (≥8 mm height + ∅ ≥ 4 mm), placed in good-quality mature bone, reaching enough primary stability (>40 Ncm), restored with a provisional prosthesis screwed for at least 6 weeks with minimal functional occlusion, and preferably splinted to other comparable implants [24,25,26,27].

Despite the previous ideal conditions for immediate loading, other suboptimal scenarios are usually found by clinicians and researchers, which led other authors to conclude in recent meta-analyses that, compared with conventional loading, immediate loading is associated with a higher incidence of implant failure [28,29]. The SSS was high enough to support the precedent statements.

### 3.4. Effect of Transmucosal Abutments

There is increasing evidence (SSS = 13) supporting the use of transmucosal abutments of at least 2 mm height for minimizing the marginal bone loss of implants [30,31,32], which is probably the best clinical indicator of implant success. In fact, it has also been reported that implants with a shorter polished collar not only do not show additional bone resorption, but also maintain higher crestal bone levels [33]. However, all the previous evidence comes from bone-level implants with conical connection, and there is a lack of studies focusing on either external hexagon connections or tissue-level implants. In this regard, it was observed in a well-performed animal study that 1.5 to 2.0 mm of bone loss occurred around bone-level, external-hexagon butt-joint implants when the abutment was connected at the stage-two surgery. However, if the abutment was placed on the implant at the time of first surgery (implant placement) or when the implant inserted is tissue-level or one-body implant, no or minimal bone loss was observed [34].

Therefore, the transmucosal abutment should be inserted on the same day of implant surgery, whenever possible, to minimize marginal bone loss and subsequent soft-tissue changes [35]. The use of transmucosal abutments avoids the repeated connection/disconnection of distinct additments at the implant platform level, hence enhancing the stability of marginal bone [36].

### 3.5. Effect of Prosthetic Fit

First, two levels should be distinguished in order to assess this relevant topic for the prognosis of dental implants. If a dental implant is considered the endoprosthesis (infrastructure in direct contact with the bone), the transmucosal abutment is the mesostructure and the prosthesis itself is the suprastructure. Then, the fit between infrastructure and mesostructure occurs at a bone-level and should ideally be hermetic (impenetrability), whereas the fit between mesostructure and suprastructure happens at the gingival level and should ideally be passive (passive fit).

There is emerging evidence that biological tolerance to inaccuracy in fit occurs mainly between the supra- and mesostructure [13] rather than between the infrastructure and the mesostructure [37]. This is because in a septic environment such as the mouth, all the micro gaps will shortly be filled by microorganisms, which may lead to gingival inflammation and bone loss depending on the distance to the gingival tissues; hence, the presence of pathogens is more threatening at the implant-abutment interface than at the abutment-prosthesis interface.

When machine-prefabricated additments are used, the fit will be much better than when technicians have to manufacture the prosthetic components, whose misfit usually ranges between 40 μm and 120 μm with distinct techniques [38]. It must be taken into account that with each manufacturing step (impression, cast pouring, wax-up, investing, metal casting, polishing, and veneer application), further inaccuracies are introduced in the final framework.

Nevertheless, if the suprastructure (technician-made prosthesis) is screwed on machine-prefabricated transepithelial abutments (at the gingival level instead of implant level) then the supra-mesostructure misfit will be brought close to the gingival margin and away from the bone. The prosthesis construction at the gingival level would therefore increase the stability of marginal bone [39]. The quality of sealing at the implant-abutment interface will affect the stability of peri-implant bone [37].

It is widely accepted that marginal discrepancies of 10 μm to 150 μm would be clinically acceptable in the long-term. However, from a biological point of view, the gap size should be smaller than any periodontally harmful bacteria (<2 μm) [40], which is technically unfeasible to date, and subsequently a certain grade of bacterial colonization at the different interphases is ineluctable [41].

Misfit between abutment-prosthesis interphases is thought to create uncontrolled strains in the prosthetic components and lead to technical complications such as screw loosening, component fractures and, at worst, loss of implants or prostheses [40].

Passive fit is the maximal spatial congruence between implant/abutment and frameworks after tightening all screws. It is difficult to assess clinically by inspection. It has been estimated that at a working distance of 25 cm, two points as close as 100 μm to each other can be distinguished as individual points by the naked eye [40]. Hence, with the help of a 2x magnification lens, a sensitivity of 50 μm can be expected at the same distance, which could be currently considered as a good fit [40]. However, visual assessment of the passive fit is difficult or even impossible for sub-gingivally positioned and conical type implants, which is another reason for working on transmucosal abutment rather than on implant level for the prosthesis construction.

In a well-performed retrospective study with a large observation period (mean: 19 years; range: 12 to 32 years) including 30 mandibular full-arch implant supported prostheses, it was found that the average misfit of patients with history of screw-related adverse events was slightly but significantly higher (169 ± 32 μm) than counterparts (134 ± 30 μm), although no linear correlation between misfit and marginal bone loss [42] was found.

To date, there is limited clinical evidence that supports the expected correlation between the degree of passivity of screw-retained prosthesis fit and the rate of mechanical and biological adverse events [40], which highlights that there might be a safe range of misfit beyond which the static forces can potentially cause both technical and biological complications [43]. The limits and the interpersonal variability of such tolerance are still unknown. In any case, this topic seemed to be adequately supported by the literature (SSS = 32.3).

### 3.6. Effect of Provisionalization

In recent years, the use of provisional restorations on implants during healing has been considered a key factor for achieving predictable outcomes in the esthetic zone Currently, the provisionalization of implants is virtually only used for full-arch rehabilitations or when replacing cosmetic teeth. A recent review describes and addresses in depth various types of implant-related interim prostheses regarding their biomechanics, support, time of loading, techniques and occlusal contact [44]. The provisional prosthesis provides patients with a quick restoration of esthetics and function, serves as a diagnostic template for the final restoration, and acts as a scaffold to guide soft tissue contour for enhanced esthetics [45].

Depending on the support, provisionals can be either removable or fixed. Removable provisionals are cheaper and easier to manufacture/adapt, but due to their insufficient stability (static and dynamic mobility), these prostheses could put the underlying healing implant/graft at risk. therefore, they should be avoided when grafting tissues or when the primary stability of implants is low. By contrast, fixed interim prostheses (either supported by adjacent teeth or by implants) provide the patient with better esthetics and comfort, which is especially valuable for those who have not experienced edentulism or removable prostheses before [44].

Even though there are no clinical trials assessing treatment outcomes with different types of provisionals, most clinicians are aware of the positive effects of the implant provisionalization for creating a more esthetically predictable definitive restoration, especially in the premaxilla [13]. However, the need for extra time, components, and cost make this intervention less common than would be recommendable in the daily practice. The scientific fundaments of provisionalization only obtained 12 points.

### 3.7. Effect of Type of Retention (Screw vs. Cement)

Dental literature has plenty of conventional and systematic reviews exploring the advantages and disadvantages of cement- vs. screw-retained implant-supported prosthesis [46,47,48,49,50]. These usually report ambivalent results that bring to light that none of the fixation methods are clearly advantageous over the other, leaving clinicians with conflicting information. Both methods have their advantages and their specific indications, making the randomization of this intervention in clinical trials unviable Therefore, the best evidence comes from meta-analyses in which both types of retention are grouped together in order to quantify the comparative risks.

The main advantage of screw-retained implant reconstruction is probably its predictable retrievability, which facilitates its removal for hygiene maintenance, repairs, or surgical interventions. On the other hand, the main advantages of cement-retained prostheses are the improved esthetics, since the morphology and position of the replacement tooth is not as conditioned by the prosthetic screw access hole. Furthermore, the ability to compensate implant position discrepancies and the easiness to achieve adjustment passivity of the suprastructure are significant advantages.

Although clinicians should be aware of the differential indications and accordingly select the most appropriate method of retention for each individual patient, they actually grow accustomed to one of the two options and resolve most of their cases with the same strategy, whose alleged superiority is essentially based on beliefs rather than on science. According to a good systematic review carried out by Wittneben et al. [51], there were no significant differences between retention types for implant survival, but the review showed more technical and biological complications for cemented prostheses. A later review of the same research group concluded that screw-retained fixed partial prostheses had a significantly higher rate of technical complications, and screw-retained full-arch prostheses demonstrated a notably high rate of veneer chipping. However, when “all fixed prostheses” were considered, significantly higher rates of technical and biological complications were seen for cement-retained prostheses [46]. In summary, a universal recommendation cannot be made for either cementation or screw retention, because each option has its own indications, but it could be argued that cemented reconstructions seem to show more serious biological complications, whereas screw-retained reconstructions present more technical problems [48,50,51]. Since most clinicians are more afraid of biological than technical complications, screw-retained prostheses are probably the preferred option for most clinicians today. This topic was supported by a high scientific score (SSS = 41).

### 3.8. Effect of Impression Techniques

At some point during treatment stage, the position of each dental implant in relation to neighboring implants or teeth, as well as the morphology of the remaining tissues, should be recorded and transferred to a working stone cast for the manufacturing of an implant-supported prosthesis. This transfer may be either at the implant connection or at the abutment. Moreover, the transfer may follow either conventional or digital guidelines.

The conventional workflow for dental implant impressions involves screw-retained impression copings that are attached to implants and impression trays loaded with elastomer impression material. Within the conventional method, the impression copings can be either retained in the cured impression material (pick-up method/direct technique) or can remain on the implants and be repositioned later in the negative track left in the impression (transfer method/indirect technique). The pick-up method needs open impression trays, whereas the transfer method is performed with closed impression trays.

Because the pick-up technique allows the impression copings to remain in the impression during setting, it reduces the deformation of the impression material on recovery from the mouth and eliminates the concern for properly replacing the coping back into its respective space in the impression [52].

For making an impression of neighboring implants (partial or full arch), especially when there is certain angulation between each other or when implants are deep sub-gingivally (>3 mm), the pick-up impression copings should be splinted to each other with a rigid material (acrylic resin, stainless steel pins, plaster) before adding impression material to obtain a reliable record [53]. The more rigid the splinting material is, the more accurate master cast will be [54].

On the other hand, digital implant impressions allow the acquisition of implant positions by connecting scan bodies to either implant or abutments in order to create an accessible surface for optical acquisition by intraoral scanning devices [55].

Although the accuracy of implant impression techniques is a topic that has been widely studied in dental literature—mainly comparing non-splinting vs. splinting techniques, direct vs. indirect techniques, parallel vs. tilted implants and diverse impression materials [55]—today most of the evidence is of low level because it does not include sufficient data from in vivo studies [56]. This is probably because in clinical settings there would be so many potential modulating factors to control that this would hamper a true comparison among techniques. Based on the accumulated experience from experimental studies, it can be concluded that, to date, pick-up techniques of rigidly splinted copings are, in general, the best choice for accurately recording the implants’ positions [55]. These statements were solidly supported by scientific evidence (SSS = 41.4).

In general, conventional implant impressions of angulated implants are significantly less accurate compared to parallel implants [57]. Digital implant impressions are as accurate as conventional implant impressions, mainly for single and partial edentulous spaces. However, for completely edentulous jaws or long edentulous distances, conventional impression techniques seem to be a more precise option nowadays [58], although promising results are emerging from recent clinical trials [59]. Furthermore, intraoral scanning proved to be more time-efficient as compared to the conventional impression technique [60]. Lastly, dental restorations made with the digital impression technique showed statistically similar marginal discrepancies in comparison with those obtained with the conventional impression technique [61].

### 3.9. Effect of Manufacturing Technique

Once the working cast is virtually or conventionally obtained, the fabrication of the prosthesis can also follow either conventional or digital workflows. The traditional process for implant-supported prostheses starts at the dental laboratory, where dental stone casts with implant analogs are poured from conventional implant impressions. Later, abutments and suprastructures are subsequently designed on the stone cast by means of a manual wax-up. Thereafter, the manufacturing process involves casting/pressing procedures based on the lost-wax technique. Once the mesostructure is manufactured and checked for passive fit intraorally, the final restoration is then covered by esthetic veneering material (ceramic, composite, resins); this is a process involving a variety of complex manual steps, materials and equipment, as well as skills and expertise on the part of dental technicians.

However, the digital pathway for the manufacture of implant-supported reconstructions may usually involve either intraoral scanning or scanning of conventionally manufactured models, computer-aided design of both interim and final reconstructions, and computer-aided manufacturing of devices/prosthesis by additive and/or subtractive techniques. In addition, a hybrid workflow of pure conventional and full digital flow is also usually adopted by clinicians. Nonetheless, in any case, the finalization of the prostheses needs technical manual interventions for staining, glazing, polishing, etc. [60].

It seems evident that the quality of the adjustment of fixed prostheses onto either implants or abutments is a key element for ensuring long complication-free prosthetic survival. In 2014, Boitelle et al. performed an exhaustive review of the marginal fit attained by several CAD-CAM technologies used for tooth-supported prostheses which concluded that CAD-CAM provided a better fit (usually < 80 μm) than conventional methods (usually > 100 μm) [62]. Additionally, a recent systematic review evaluating the advantages of digital technologies for the manufacturing of implant-supported rehabilitations reported an increased time efficiency of CAD/CAM systems as compared to conventional fabrication procedures [60] when the following guidelines were adopted: model-free fabrication and use of monolithic ceramic single crowns bonded to prefabricated abutments. Moreover, the abovementioned conclusions were based solely on the experience of Tim Joda and Urs Brägger’s studies with posterior single monolithic crowns [63,64,65], since there is a lack of studies focusing on partial and full rehabilitations, as well as on the testing of other fabrication methods or materials. Furthermore, unlike standardized abutments, customized abutments require more time in the dental laboratory. In addition, the reviewed studies did not report on the waiting time during the fabrication process, which ultimately is an important parameter for both clinicians and patients. In any case, the main rationale for using digital methods should rely on precision rather than on speed of the manufactured prosthesis.

When three common manufacturing techniques were compared (milled, laser-sintered, and casted) in terms of microroughness of the mating surfaces, it was found that milled structures were significantly smoother (29 μm on average) than laser-sintered (115 μm) or casted components (98 μm) [66]. Similarly, the magnitude of the marginal micro gap was found to be significantly smaller within milled abutments (0.73 μm), than the gap of sintered (11.30 μm) and cast (9.09 μm) abutments [66]. The linear correlation between microroughness and micro gap was found to be huge (*r* = 0.96).

Nevertheless, recent evidence points out that conventional manufacturing techniques could still be considered as effective and reliable for the fabrication of precise prostheses as CAD-CAM technologies [67], delivering marginal discrepancies bellow 120 μm, which is a clinically acceptable threshold. This topic was also adequately supported by the literature (SSS = 33.1). Regarding the effect of the restorative material used in the implant-supported prostheses, two recent meta-analyses stated that the prosthetic material selection has no clinically relevant influence on mid- and long-term survival of implants and prostheses [68,69].

### 3.10. Effect of Occlusal Considerations

This is probably the most interesting but conflicting topic, because despite the huge quantity of papers reporting on occlusal considerations of implant therapy, there is insufficient evidence (poor study design, diverse biases, ambiguous results) to establish firm clinical guidelines for implant occlusion [68]. Thus, the most common approach is to apply principles and methods deriving from conventional prosthodontics, considering the role of overloading factors in both biological and technical complications. In this regard, it is advisable to review the work of Kim et al. [69] for better understanding the rationale of distinct occlusal strategies in diverse situations.

We are well aware that the following guidelines are extrapolated from conventional prosthodontics and thus are not evidence-based (SSS < 10), but at least are aimed at reducing excessive load in an attempt to minimize the potential harmful effects on the system. Therefore, the following directives seem advisable:Using the occlusal scheme mutually protected whenever possible.Trying to avoid, whenever possible, non-axial loading of implant-borne (mainly for single restorations), although the natural disposition of teeth and the resultant forces of occlusion during mastication are rarely axial.Fabricating low cusp inclinations and fitting the occlusion with shim stock clearance at intercuspal position and centric occlusion.Giving low prominence to the implant-prostheses during mandibular excursions. Nevertheless, probably thanks to the great neurophysiological resilience of the masticatory system, a gradual adaptation to subtle and gross changes in the occlusal status is usually observed.

## 4. Discussion

The search strategy used in this study was not exhaustive enough to be named as systematic but, at least, for each risk factor the most appropriate/updated articles were selected, giving preference to systematic reviews, meta-analysis, and randomized clinical trials.

This review has highlighted that the best scientifically grounded prognostic factor is the effect of loading time (SSS > 50 in Figure 3), followed by other factors moderately supported by the literature (SSS ranging between 30 and 50 in Figure 3) such as impression techniques, type of retention, manufacturing techniques, and prosthetic fit. However, to date, the effect of the use of transmucosal abutments, provisionalization, and mainly the occlusal scheme on implants are not properly based on high scientific evidence, and future research should be directed towards these prosthodontics topics. Table 1 shows the papers used to support the evidence among the reviewed prosthetic factors.

According to the works analyzed, this review draws the following conclusions: External connection implants involve more complications than morse taper connection implants. Immediate loading is safe when the following conditions are met: implants are placed in mature bone of good quality; they reach enough primary stability (>40 Ncm); implants are restored with a provisional screwed prosthesis for at least six weeks with minimal functional occlusion; and they are preferably splinted to other comparable implants. Transmucosal abutment ≥ 2 mm should be connected to the implant on the same day of surgery. Cemented prostheses seem to cause more serious biological complications, whereas screw-retained reconstructions show more technical problems. For impressions, the pick-up technique of rigidly splinted copings is probably the best choice for accurately recording implants’ positions. Milled suprastructures are the best option for accurately manufacturing fitted prostheses. The restorative material seems to have no significant influence on clinical performance. The occlusal scheme should be focused on protecting implants from overloading.

## 5. Conclusions

The highest scientific base (SSS = 60) was found for the effect of loading time on the implant prognosis, mainly because this factor is easy to randomize in clinical settings and therefore it has been supported by better study designs. However, the current literature provides insufficient evidence for most of the investigated topics.

## Figures and Tables

**Figure 1 jcm-10-00816-f001:**
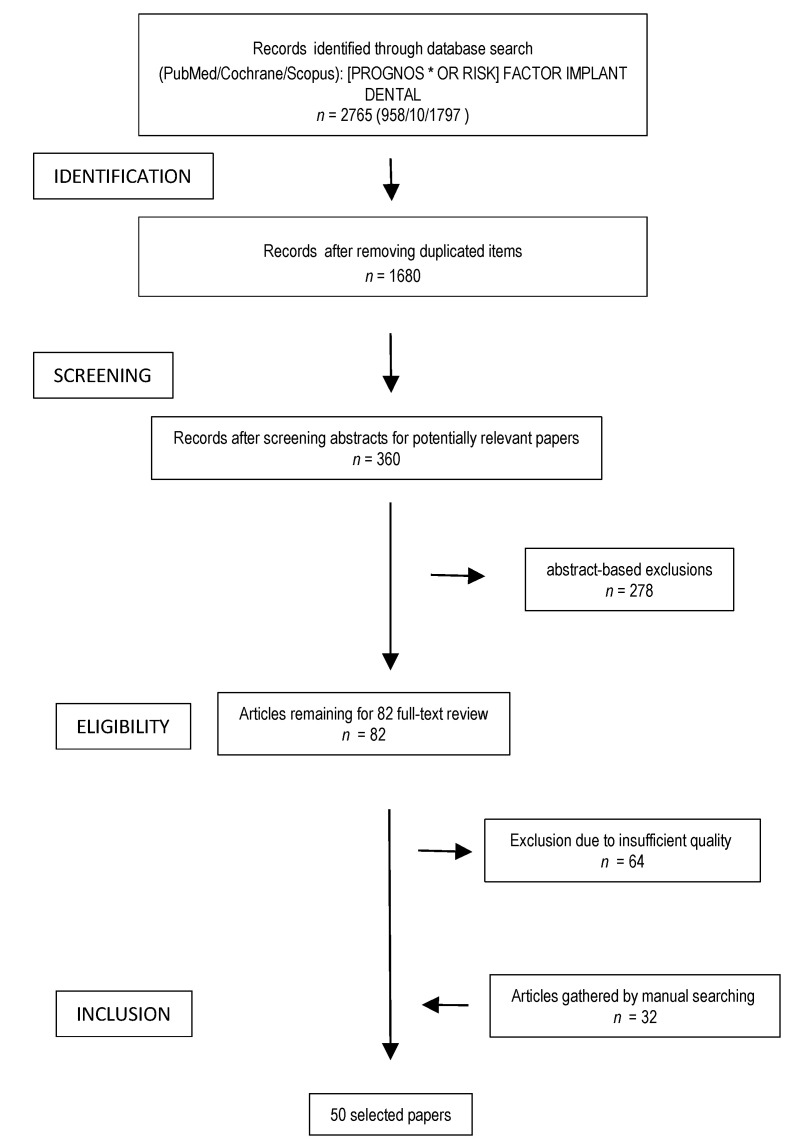
Flow chart of the search strategy of this comprehensive review.

**Figure 2 jcm-10-00816-f002:**
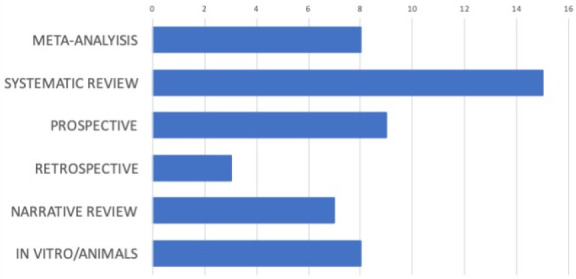
Design of the selected papers.

**Figure 3 jcm-10-00816-f003:**
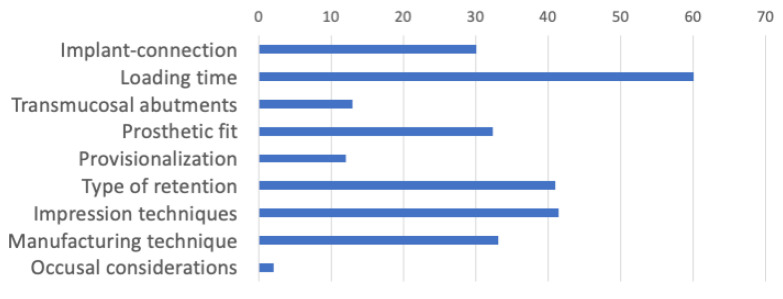
Comparison of the Summary Scientific Scores (SSS) among distinct prognosis factors.

**Table 1 jcm-10-00816-t001:** Summary and brief description of the included studies supporting each pronostic factor.

TOPIC	First Author (Year) [Reference]	Type of Study	Main Focus	Quality Level
IMPLANT CONNECTION				
1	Vetromilla (2019) [21]	Systematic Review	Mechanical and biological effect on premaxilla	++
2	Caricasulo (2018) [22]	Meta-analysis	Marginal bone loss	+++
3	Goiato (2015) [23]	Systematic Review	Mechanical, biological and esthetic performance	+++
LOADING TIME				
4	Esposito (2007) [24]	Cochrane Review	Effectiveness of immediate, early or conventional loading on implants	+++
5	Esposito (2013) [25]	Cochrane Review	Effectiveness of immediate, early or conventional loading on implants	+++
6	Gallucci (2009) [26]	Systematic Review	Effectiveness of distinct loading moments on distinct clinical conditions	+++
7	Papaspyridakos (2009) [27]	Meta-analysis	Effectiveness of distinct loading moments for edentulous jaws with fixed prostheses	+++
8	Chen (2019) [28]	Meta-analysis	Effectiveness of distinct loading moments for edentulous jaws with fixed prostheses	+++
9	Pardal-Peláez(2020) [29]	Meta-analysis	Effectiveness of immediate loading on marginal bone loss	+++
TRANSMUCOSAL ABUTMENTS				
10	Chen (2019) [30]	Meta-analysis	Abutment heigh and marginal bone loss	+++
11	Galindo-Moreno (2016) [31]	Clinical Trial	Abutment heigh and marginal bone loss	+
12	Blanco (2018) [32]	Clinical Trial	Abutment heigh and marginal bone loss	++
13	Tan (2011) [33]	Clinical Trial	Implant-neck and hard/soft tissues	++
14	Hermann (2001)	Animal Study	Biological width around one and two-piece titanium implants	++
15	Wang (2017) [35]	Meta-analysis	One-time vs. repeated abutment connections in platform-swithed implants	+++
16	Koutouzis (2017) [36]	Meta-analysis	Repeated abutment connections and marginal bone loss	+++
Prosthetic Fit				
17	Lewis (2011) [13]	Narrative Review	Prosthodontic considerations for optimizing outcomes for single-implants	+++
18	Sasada (2017) [37]	Narrative Review	Biological Consequences of distinct type of implant abutment connections	++
19	de Luna Gomes (2019) [38]	In vitro	Misfit of frameworks made by distinct techniques	+
20	Hernández-Marcos (2018) [39]	Clinical Trial	Marginal bone loss around implant- vs. abutment-level restorations	+
21	Katsoulis (2017) [40]	Systematic Review	Misfit and clinical performance	++
22	Abduo (2014) [41]	Narrative Review	Fit of CAD-CAM frameworks	++
23	Jokstad (2015) [42]	Clinical Trial	Long-term clinical effects of misfit in full-arch prostheses	++
24	Jemt (1996) [43]	Clinical Trial	Assesment of the precision of fit	+
Provisionalization				
25	Siadat (2017) [44]	Narrative Review	Provisional prostheses options	+
26	Santosa (2007) [45]	Narrative Review	Provisional prostheses options	+
Screw vs. Cement				
27	Millen (2015) [46]	Systematic Review	Complication rates with fixed prostheses	+++
28	Wismeijer (2014) [47]	Narrative Review	Consensus staments on implant dentistry	+
29	Sailer (2012) [48]	Systematic Review	Survival and complication rates	+++
30	Lemos (2016) [49]	Meta-analysis	Marginal Bone Loss	++
31	Gaddale (2020) [50]	Meta-analysis	Incidences of Complications	+++
32	Whittneben (2014) [51]	Systematic Review	Clinical performance	+++
Impression Techniques				
33	Del’Acqua (2010) [52]	In vitro	Accuracy of two impression techniques	+
34	Martinez-Rus (2013) [53]	In vitro	Accuracy with different angulations and subgingival levels	++
35	Kim (2015) [54]	Systematic Review	Dimensional Accuracy	+++
36	Del’Acqua (2010) [55]	In vitro	Splinting material rigidity	+
37	Flügge (2016) [56]	In vitro	Digitalization with intraoral scanners	++
38	Flügge (2018) [57]	Meta-analysis	Accuracy among distinct condition	+++
39	Schimidt (2020) [58]	Clinical Trial	Digital vs. Conventional full arch impressions	+
40	Cappare (2019) [59]	Clinical Trial	Digital vs. Conventional full arch impressions	+++
41	Mühlemann (2018) [60]	Systematic Review	Efficacy and effectiveness of digital vs. conventional techniques	+++
42	Chochlidakis (2016) [61]	Meta-analysis	Digital vs. conventional in fixed prosthodontics	+++
Manufacturing Technique				
43	Boitelle (2014) [62]	Meta-analysis	Fit of CAD-CAM restorations	+++
44	Joda (2015) [63]	Clinica Trial	Time-Efficiency Analysis of digital vs. conventional workflow in single implants	+
45	Joda (2016) [64]	Clinica Trial	Time-Efficiency Analysis of monolithic single implants crowns	++
46	Joda (2017) [65]	Clinica Trial	Time-Efficiency Analysis of monolithic single implants crowns	+++
47	Fernández (2014) [66]	In vitro	Microroughness and microgap of three tehcniques	+
48	Papadiochou (2018) [67]	Systematic Review	Marginal fit depending on the restorative material and fabrication techniques	+++
Occlusal considrations				
49	Koyano (2015) [70]	Narrative Review	Clinical Guidelines	++
50	Kim (2005) [71]	Narrative Review	Clinical Guidelines	++

The quality of the distinct studies was categorized as + (low), ++ (medium), +++ (high) depending on the items that they fulfill depending on the study design: For Systematic Review, the quality was computed by totaling three dichotomic questions: Was a comprehensive literature search performed (at least two databases)?; Was the scientific quality of the included studies assessed and documented?; Were the methods used to combine or summarize the findings of studies appropriate? For Meta-analysis, the quality was computed by totaling the following dichotomic questions: Was there enough quantity and homogeneity within the included studies?; Was the scientific quality of the included studies assessed and documented?; Were the methods used to combine or summarize the findings of studies appropriate? For Clinical Trials, the quality was computed by totaling the following dichotomic questions: Was the intervention randomized and concealed?; Was the sample size greater than 30 participants?; Were patients followed for more than one year? For narrative reviews, the quality was computed by totaling the following dichotomic questions: Was the PICO question properly described?; Was the hierarchy of evidence properly described?; Are the findings of clinical relevance? For the animal and in vitro studies, the quality was computed by totaling the following dichotomic questions: Was the design of the experiment effective enough to fulfil primary objectives? Were the analytical methods appropriate for the primary purpose? Are the findings of clinical relevance?

## Data Availability

The data presented in this study are available in the article.

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
