# Peer review of "A Review of the Major Prosthetic Factors Influencing the Prognosis of Implant Prosthodontics"

_jcm, 2021, doi:10.3390/jcm10040816_

Round 1
Reviewer 1 Report
Dear author,
this systematic review is very interesting and original in its intent.
Has registration been made in the PROSPERO systematic review register? If this has not been done, we recommend that you do so now and include the provisional registration number in the text of the article.
The number of titles that led to the initial search is not reported either in the flow chart or in the text.
I would suggest, for greater clarity of presentation, to add a summary table of the included studies and the parameters analyzed for each study.
The "Effect of transmucosal abutments" parameter should be further discussed in my opinion, including and citing the conclusions of this recent and interesting systematic review: PMID: 32956437
Best regards
Author Response
REVIEWER 1
Dear author,
this systematic review is very interesting and original in its intent.
Has registration been made in the PROSPERO systematic review register? If this has not been done, we recommend that you do so now and include the provisional registration number in the text of the article.
Reply: This review has now been submitted to PROSPERO with the following provisional registration number 233198, but the complete revision of the proposal will take at least 28 days.
The number of titles that led to the initial search is not reported either in the flow chart or in the text.
Reply: Done
I would suggest, for greater clarity of presentation, to add a summary table of the included studies and the parameters analyzed for each study.
Reply: According to your suggestion a summary table is now included as Table 1.
The "Effect of transmucosal abutments" parameter should be further discussed in my opinion, including and citing the conclusions of this recent and interesting systematic review: PMID: 32956437
Reply: I have just read this highly relevant work, but unfortunately it was not available when this search was carried out (April, 2020).
Best regards
Reviewer 2 Report
The paper is of interest and well written. Prosthodontic factors are sometimes unduly neglected in dental implant rehabilitation but are fundamental for the survival and the success of the rehabilitation. As a consequence, reviews on this topic are the welcome.
However, the methodology followed for the systematic review presents several flaws. The topic addressed by the author is very broad and the methodology applied is not sufficient to answer the research question. The search strategy should be reconsidered as the present strategy failed to find many pertinent and important papers. This is also demonstrated by the fact that only 18 papers have been selected from the original search and even as much as 32 papers have been retrieved by manual searching of the reference list of the 18 papers.
As a consequence, many pertinent papers have not been included. I herein list some important papers on the topic that have not been considered in the present review:
- Paolo Pesce, Luigi Canullo, Maria Gabriella Grusovin, Hugo de Bruyn, Jan Cosyn, Paolo Pera. Systematic review of some prosthetic risk factors for periimplantitis. J Prosthet Dent 2015; 114(3):346-50
- Menini M, Setti P, Pera F, Pera P, Pesce P. Accuracy of multi-unit implant impression: traditional techniques versus a digital procedure. Clin Oral Investig 2018;22:1253-1262.
- Pera P, Menini M, Pesce P, Bevilacqua M, Pera F, Tealdo T. Immediate versus delayed loading of dental implants supporting fixed full-arch maxillary prostheses: a 10-year follow-up report. Int J Prosthodont 2019;32:27-31.
On the other side the author does not present any inclusion/exclusion criteria that might help increasing the quality of included papers and the quality assessment method applied was not sufficient to assess the risk of bias.
In addition, the author has failed to consider all the factors affecting the prognosis of implant prosthodontics i.e. considerations regarding the restorative material have been omitted.
In the end, given such limitations, the conclusions reported by the authors are not sufficiently supported.
I suggest narrowing the research making it more focused and strengthening the methodology applied or in alternative presenting the manuscript as a narrative review instead of a systematic review.
Figure 1: please report how many papers have been retrieved for each database.
Author Response
28 January, 2021
Dear Reviewer:
Thank you for your revision (time and insights). All your comments have been addressed in order to improve the manuscript. The parts of the manuscript that have been modified according to your suggestions are highlighted in blue throughout the text. A point-by-point response to your comments is detailed below consecutively. I hope all these modifications will be finally satisfactory.
REVIEWER 2
The paper is of interest and well written. Prosthodontic factors are sometimes unduly neglected in dental implant rehabilitation but are fundamental for the survival and the success of the rehabilitation. As a consequence, reviews on this topic are the welcome.
- However, the methodology followed for the systematic review presents several flaws. The topic addressed by the author is very broad and the methodology applied is not sufficient to answer the research question.
Reply: We agree with you, and accordingly the focus of this study has been redirected towards assessing the level of evidence supporting the major prosthetic factors affecting the prognosis of implant-prosthodontics. We are well aware that this is a broad topic but this paper was deliberately intended to make a comprehensive review of the best scientificly grounded prognosis factors in a try to give clinicians a simplified but panoramic explanation of the main prosthetic determinants in implant prognosis. In fact this broad-focus methodology allowed the identification of the folowing factors implant-connection, loading protocol, inteface integration, provisionalization, type of retention, impression technique, fabrication technique, biomechanics, occlusion. In addition it could be argued that the methodology used here (computing the scientific score as an ordinal estimation) could be useful when no comparable data is availble for statistical analysis.
It is frequently reported that because of the heterogeneity of the included studies and the small number of randomized clinical trials available in the literature, the primary outcome of the systematic review and meta-analysis can not be fulfilled.
- The search strategy should be reconsidered as the present strategy failed to find many pertinent and important papers. This is also demonstrated by the fact that only 18 papers have been selected from the original search and even as much as 32 papers have been retrieved by manual searching of the reference list of the 18 papers.
As a consequence, many pertinent papers have not been included. I herein list some important papers on the topic that have not been considered in the present review:
- Paolo Pesce, Luigi Canullo, Maria Gabriella Grusovin, Hugo de Bruyn, Jan Cosyn, Paolo Pera. Systematic review of some prosthetic risk factors for periimplantitis. J Prosthet Dent 2015; 114(3):346-50
- Menini M, Setti P, Pera F, Pera P, Pesce P. Accuracy of multi-unit implant impression: traditional techniques versus a digital procedure. Clin Oral Investig 2018;22:1253-1262.
- Pera P, Menini M, Pesce P, Bevilacqua M, Pera F, Tealdo T. Immediate versus delayed loading of dental implants supporting fixed full-arch maxillary prostheses: a 10-year follow-up report. Int J Prosthodont 2019;32:27-31.
Reply: This is a unavoidable consequence of having a wide focus and a lack of studies addressing the impact of prosthodontic, biomechanical and occlusal factors on the prognosis of implant prosthetics. The complementary search was oriented to capture the best/updated evidence underying each of the previously identified prognosis factors. That the reason of this aparently ineffectivestrategy. It has been acknowledged in Methods and discussed latter that the search strategy was not exhaustive enough to be named as systematic but, at least, for each factor the most appropriate/updated articles were selected, giving preference to systematic reviews, meta-analysis and randomized clinical trials.
As a result the findings reported in the systematic review of Pesce et al 2015 is actually included in Canullo et al 2016 which has already been reviewed in this work: i.e. [Ref 11 in this study]. However the study of Menini et al study 2018, was probably discarded because it is a in vitro study addressing a question that has already been addressed by studies with better designs such as Schmidt et al 2020 in a clinical setting [Ref 58] and, for example, Chochlidakis 2016 in a meta analysis [Ref 61] within the same topic. Similarly the prospective cohort study of Pera 2019 was dicarded because the best evidence for this topic was obtained from Cochrane reviews [Ref 24 and Ref 25] and a recent meta-analysis of randomized controlled trials [Ref 28].
- On the other side the author does not present any inclusion/exclusion criteria that might help increasing the quality of included papers and the quality assessment method applied was not sufficient to assess the risk of bias.
Reply: We have described in M&M the inclusion/exclusion criteria to clarify the quality assement method (Also described in Table 1 footnote)
- In addition, the author has failed to consider all the factors affecting the prognosis of implant prosthodontics i.e. considerations regarding the restorative material have been omitted.
Reply: It has been included in results the following statement “regarding the effect of the restorative material used in the implant-supported prostheses two recent meta-analyses stated that the prosthetic material selection has no clinically relevant influence on mid- and long-term survival of implants and prostheses [68,69].
- In the end, given such limitations, the conclusions reported by the authors are not sufficiently supported.
Reply: The limitations underlying each prosthetic factors have now been properly discussed and the final conclusion, based on the findings of the study has been rewriten as follows “The highest scientific base (SSS=60) was found for the effect of loading time on the implant prognosis, mainly because this factor is easy to randomize in clinical settings and therefore it has been supported by better study designs. However the current literature provides insufficient evidence for most of the investigated topics.”
- I suggest narrowing the research making it more focused and strengthening the methodology applied or in alternative presenting the manuscript as a narrative review instead of a systematic review.
Reply: Given the fact that the purpose of this study is to give journal readers a panoramic view of the main prosthetic determinants of the prognosis of implant-supported prostheses, the paper has been presented as a narrative review and the title has been changed accordingly.
Figure 1: please report how many papers have been retrieved for each database.
Reply: Done
Round 2
Reviewer 2 Report
The papers was correctly revised and it seems appropriate to publish in the actual version